# Early Critical Thinking in a Mandarin-Speaking Child: An Exploratory Case Study

Xuemei Shao [1,*], Ruying Qi [1], Satomi Kawaguchi [1] and Hui Li [2,3]

1 Bilingual Research Lab, School of Humanities and Communication Arts, Western Sydney University, Sydney, NSW 2116, Australia; r.qi@westernsydney.edu.au (R.Q.); s.kawaguchi@westernsydney.edu.au (S.K.)
2 Shanghai Institute of Early Childhood Education, Shanghai Normal University, Shanghai 200234, China; philip.li@mq.edu.au
3 School of Education, Macquarie University, Sydney, NSW 2109, Australia
* Correspondence: 90947816@westernsydney.edu.au

**Abstract:** Critical thinking in children is a growing concern for early childhood educators; however, few studies have examined children's critical thinking in an out-of-class context. This case study aimed toward filling this research gap by examining the critical thinking of a Mandarin-speaking child aged 5 years and 8 months in an out-of-class context. The child's natural utterances produced in free conversation and story-readings have been audio- and video-taped twice a week over two months. The recordings have been transcribed and analyzed according to the Delphi Report and 'level of questions' to examine the child's critical thinking level. Findings revealed that the child demonstrated critical thinking, and two indicators, 'spontaneous statements' and 'continuous questions', reflected children's critical thinking level. It also found that these categories were reasonable and practical to identify young children's critical thinking levels.

**Keywords:** critical thinking; preschool children; critical thinking level; Mandarin-speaking; case study

## 1. Introduction

For over a century, educators and philosophers have highly valued critical thinking [1–6]. For the past few decades, many educators have dedicated themselves to cultivating children's critical thinking. Due to the young children's innate desire, inborn and natural curiosity, early childhood has always been considered a critical period to foster children's critical thinking [7]. Therefore, educators highly recommend that developing children's critical thinking should be and is essential to be integrated into the curricula of early childhood classrooms [8].

Piaget, an influential psychologist, and epistemologist on child cognitive development, investigated how children develop intellectually during the early years. He identified that children go through an invariant sequence of four stages in thinking and reasoning: sensorimotor stage (infancy, birth to 2 years); preoperational stage (toddler and early childhood; age 2–7); concrete operational stage (elementary and early adolescence; age 7–11); formal operational stage (adolescence and adulthood; age 12 and up) [9,10]. He emphasized the importance of education in fostering their cognitive development. According to Piaget, children's thoughts and communications in the preoperational stage are egocentric [11]. However, the following researchers have found that children have largely lost their egocentric thinking by four years of age because they can think from others' perspectives [12,13]. From the view of psycholinguistics, Levelt stressed that thought and language are directly related since the message represents the speaker's prepositional language of thought [14]. Therefore, children's language productions, such as statements, reasoning, and questioning, are valuable indicators of their cognitive levels and knowledge of the world.

Although much research has been conducted on children's critical thinking, most has been conducted in a classroom context with interventional instruction activities, i.e., exam-

ining the change of children's critical thinking levels before and after specific interventions. Interestingly, children's critical thinking in out-of-class settings has been barely discussed so far, making it difficult to understand young children's critical thinking comprehensively. However, out-of-class settings have also been considered significant in children's early life and real-life application.

To address these research gaps, the current case study endeavors to examine a Mandarin-speaking child's critical thinking in an out-of-class setting with the following research questions:

(1) Does a preschool child show signs of possessing critical thinking in an out-of-class home setting?

(2) If yes, then what level of critical thinking has the child attained?

## 2. Literature Review

This section reviews the research on children's critical thinking below fourth grade. Three themes that are relevant to the study will be the focus of this review: critical thinking definition, critical thinking cultivation, and critical thinking development in the classroom and home environments:

### 2.1. Critical Thinking Definition

The concept of critical thinking was first proposed by Dewey [15], and since then, scholars have proposed various definitions from their own perspectives. These include Siegel [6], Lipman [16], Norris and Ennis [17], and Facione [3].

Dewey defined critical thinking as an 'active, persistent and careful consideration of a belief or supposed form of knowledge in the light of the grounds which support it and the further conclusions to which it tends' [18] (p. 6) [19] (p. 9). The whole process is also what Dewey proposed as 'reflective thinking.'

To Siegel, reasoning is viewed as a unique and necessary characteristic when thinking critically, and a critical thinker is the one who is 'appropriately moved by reasons.' From Siegel's perspective, a critical thinker's actions, judgments, and evaluations are based on rational thinking [6] (p. 32).

Lipman claimed that 'critical thinking may be defined as skillful, responsible thinking that facilitates good judgment because (1) it relies upon criteria; (2) it is self-correcting, and (3) it is sensitive to context' [16] (p. 39). According to his definition, critical thinking's outcome is good judgment. Moreover, relying on criteria, self-correction, and context-sensitivity provides three indicators to determine whether certain thinking is skillful and responsible thinking and, therefore, whether it can be classified as critical thinking.

Norris and Ennis argued that 'critical thinking is reasonable, reflective thinking that is focused on deciding what to believe or do [17].' This definition integrates both Dewey's and Siegel's definitions of reflective and reasonable thinking. However, Norris and Ennis's definition provides a more practical aim of critical thinking: 'deciding what to believe or do.'

From the literature, it is obvious that the definition of critical thinking has not reached a consensus. However, the definitions above share two essential concepts: (1) the outcome of critical thinking: good belief and good behavior; (2) the process of critical thinking: the reflective and evaluative process in which cognitive skills are applied and dispositions of critical thinking are manifested.

In 1988, the American Philosophic Association (APA) held a meeting where 46 experts in critical thinking instruction, assessment, or theory, from the fields of Philosophy, Education, and Science from the United States and Canada, came together and attempted to reach a consensus on critical thinking. One significant outcome of this meeting is the Delphi Report, in which Facione defined critical thinking into two aspects: critical thinking skills and critical thinking disposition. In addition, the Delphi experts agreed on six cognitive skills that critical thinking includes and 14 dispositions that an ideal critical thinker should possess [3]. This report has been taken as a landmark in the field, and it was adopted by

The National Research Council of the United States in 2012. More details of the Delphi Report are presented in Section 3.3.1 in conjunction with our research methods.

*2.2. Critical Thinking Cultivation*

'Philosophy for Children (P4C)' was created by Lipman in 1972. 'Community of inquiry' is the core strategy applied to initiate philosophical dialogue among children where critical thinking is expected to be developed [5,20]. P4C proves these three points: (i) children can demonstrate critical thinking; (ii) critical thinking can be taught in early childhood; (iii) children's early years could be an ideal period for educators to help them develop this significant competence in critical thinking.

Paul et al. illustrated the principles, methods, and strategies to develop children's critical thinking from K-3 to 4th–6th grades [21,22]. In addition, they provided a guide with ample examples for teachers at corresponding levels to remodel any formal lessons and practice plans into critical thinking-oriented ones. These are useful resources for critical thinking teaching in classrooms.

P4C aims to set up philosophy courses within the current curriculum in which Lipman's teaching materials and methodology are taught and conducted. Although Paul et al. do not have teaching materials, teachers' methods and frameworks are provided so that teachers can widely apply them to any course materials [23]. In other words, Lipman focuses on adding a critical thinking course to the current curriculum, while Paul et al. aim at integrating critical thinking cultivation into the existing courses. Along with Paul et al., Quinn also provided invaluable resources for teachers in helping students practice critical thinking in the classroom setting of kindergartens, primary, and early secondary schools [24]. Other researchers have also made valuable contributions to cultivating children's critical thinking by applying strategies in the classroom.

For example, Sundararajan et al. applied a quantitative method of pre-post assessment and a qualitative strand of a case study to investigate the effect of an instructional strategy: collaborative concept mapping on developing kindergarten learners' critical thinking skills of analysis and interpretation (children's age are at 5 to 6 years old on average). They provided empirical evidence to support that collaborative concept mapping could be considered an instructional strategy that teachers can use to cultivate young learners' critical thinking skills [25].

Norato Cerón investigated the impact of reading stories using an interactive reading aloud strategy on children's reading, critical thinking, and foreign language development at ages 7 to 12. Grounded theory was applied to analyze the transcribed audio recordings, and surveys were also conducted. Results proved positive effects on children's critical thinking and language abilities through reading stories aloud [26].

Ben Maad investigated the effect of another approach of 'Awakening to Language' (AtL) on critical thinking promotion on children ages 6 to 7 through a criterion variable: cultural stereotyping. 'AtL' is an approach that exposes monolingual children to foreign language environments to promote their value of openness to diversity. An intercultural stereotyping questionnaire was conducted orally by researchers with groups of no more than three children. An in-depth interview was also conducted. After the pedagogical intervention of AtL for 16 weeks, children displayed a reduction, or at least challenging, of stereotypes within and across their cultures [27].

Harbi conducted an empirical study examining the effects of comic books on educating children's (ages 10 to 13) critical thinking and ability to understand complex and abstract concepts. A qualitative method was employed to analyze discussions and written essays collected in the classroom. The research suggests that comic books could be a valuable resource for educators to practice critical thinking with their students [28]. Another empirical study was conducted by Gasparatou et al. [29]. It aimed to examine P4C in Greek kindergartens. They applied the qualitative method to compare two groups of children (5 to 6 years old) in different kindergartens by analyzing the marker-words (e.g., because, why, hence, since) as evidence of critical thinking in the transcription of the recorded

videos. Their research verified that P4C could be combined in the current Greek kindergarten curriculum; children even at the kindergarten level can benefit from P4C for critical thinking development.

*2.3. Critical Thinking Development in the Classroom and Home Environments*

Children's critical thinking development happens both in-class and out-of-class settings. However, empirical studies on this topic have concentrated on the classroom setting. Moreover, most studies utilized qualitative methods, especially when investigating young-aged children. This may be due to literacy difficulties for children to complete critical thinking assessments tests at young ages.

One example of a qualitative classroom study is conducted by Daniel and Gagnon [30]. They analyzed the transcripts of students' speech exchanges at 4 to 12 years old from 17 P4C classrooms to model pupils' critical thinking development. Based on their study, a model of the dynamic route of critical thinking development was proposed. However, although the model contributes a theoretical framework for the critical thinking development stage, it is not easy for the following researchers to employ it in transcription analysis, as the boundary between the stages is unclear.

Concerning the resources for promoting children's critical thinking, Roche introduced theories of critical thinking and the 'Book Talk' framework, proposing the 'Critical Thinking and Book Talk' methods in both the classroom and the home settings [31]. Moreover, Polette highly recommended picture books as important and favorable instruments for fostering children's critical thinking. Reflecting on her recommendation, Polette further extended the cultivation role from the classroom to librarians, parents, and homeschoolers [32]. Teachers, parents, and families also play crucial roles in children's critical thinking because the home environment provides relaxing and supportive settings to express themselves in a more intuitive, unrestricted, and honest way. Therefore, research should not be limited to within the classroom; out-of-class investigation will add invaluable insights into children's critical thinking development.

In summary, empirical studies on young children's critical thinking development are examined mainly in the classroom environment. However, the question remains: what about critical thinking development in a home environment? As a guide or 'main stimulus,' teachers play an essential role in the whole process, and the classroom is a precious setting where critical thinking will occur. However, critical thinking happens anywhere and anytime in real life. Therefore, it may not be appropriate to assume that critical thinking is only the teachers' responsibility and only happens in the classroom. However, very little research has been conducted outside the classroom setting. Therefore, we have conducted a case study to examine a child's critical thinking in out-of-class settings.

## 3. Methods

*3.1. Participant and Context*

The study's informant is a Mandarin-speaking child (Jay, fictitious name). When the data collection started, he was age 5;8 (5 years and 8 months old). He is a healthy child without any mental or physical disability. Jay lived in China; all family members are Chinese; Mandarin is their dominant language.

At age 5;4, Jay moved to Australia with his parents. Two months later at 5;6, he attended a local kindergarten. Although he had been exposed to an English-speaking environment for two months when the data collection started, he still could not produce any English except greetings and some words at the time. Moreover, the dominant language at home was Mandarin. For this reason, we collected only Mandarin language data to examine his critical thinking level, as this is the language he was able to express his thoughts freely.

The family in the study consists of a father, mother, and son. This family's socio-economic situation could be classified as middle-class, and both of his parents possess a

postgraduate degree. The parents opted to participate in this research as they are keen to learn about their son's critical thinking.

### 3.2. Research Design and Data Collection

This study adopts a case study, utilizing a naturalistic home environment, and the child's language analysis was conducted. This data collection lasted eight weeks and was performed twice per week. The child's speech productions were recorded when he was engaged in free talking, story reading, and playing with his parents in a home setting. In addition, the child's utterances were collected through video and audio recordings. The recorded data set was used for analysis in the study (details of language production samples are presented in the finding section, labelled as 'the data profile'). The child's language samples were selected randomly for analysis without specific selection criteria to ensure the data's arbitrariness.

### 3.3. Data Analysis Instrument

The child's critical thinking level was identified by analyzing his speech productions. The reasons for not using a published test to measure the child's level of critical thinking are as follows. Ennis evaluated published tests to evaluate children's thoughts and noted that only a few tests took critical thinking as their primary concern [33]. Moreover, 'none (of these tests) exist for students below fourth grade' [33] (p. 181). Ennis further states that, when probing critical thinking, the adoption of naturalistic observation, such as a case study, could be a particularly valid methodology [33].

#### 3.3.1. The Delphi Report (Facione 1990)

We utilized the Delphi Report (henceforth, in this article, abbreviated the DR) to identify the child's level of critical thinking. The report indicates that critical thinking includes six skills and 16 sub-skills (see Table 1). This study applies the DR as the instrument for the following reasons. Firstly, the DR has received broad consensus in philosophy, science, and education. Therefore, it is an objective criterion when examining critical thinking. Secondly, there are detailed descriptions of each cognitive skill with examples, and researchers can easily apply it as a criterion for language evaluation. Therefore, the data were transcribed and then analyzed using this theoretical framework. It is expected that research question 1 could be answered through the analysis.

**Table 1.** The Delphi Report.

| Skill | Sub-Skills Categorization |
|---|---|
| 1. Interpretation | Decoding significance |
| | Clarifying meaning |
| | Examining ideas |
| 2. Analysis | Identifying arguments |
| | Analyzing arguments |
| 3. Evaluation | Assessing claims |
| | Assessing arguments |
| | Querying evidence |
| 4. Inference | Conjecturing alternatives |
| | Drawing conclusions |
| | Stating results |
| 5. Explanation | Justifying procedures |
| | Presenting arguments |
| 6. Self-regulation | Self-examination |
| | Self-correction |

### 3.3.2. Questions Proposed by the Child

We also analyzed the child's question constructions, since asking questions has been regarded as a vital element in critical thinking. As Elder and Paul argued: 'The quality of our thinking is given in the quality of our questions [34].' Clasen and Bonk further emphasized that the level of thinking is directly proportional to the level of questions asked [35]. Accordingly, in this research, the questions the child proposed could be considered a reliable indicator to demonstrate his thinking level.

Bloom's taxonomy is a widely recognizable classification that categorizes the educational objectives into six hierarchical levels: 'evaluation', 'synthesis', and 'analysis' belong to the high-level thinking, while 'application', 'comprehension', and 'knowledge' belong to the low-level thinking [36,37]. In addition, Brown classified the level of questions based on Bloom's taxonomy [38]. Questions regarding thinking levels can then be summarized in the following Table 2.

**Table 2.** Level of questions.

| Thinking Level | Level of Questions | Definition | Common Question Words |
|---|---|---|---|
| Low level thinking | Knowledge questions | Eliciting factual answers and testing recall and recognition of information | Who? What? Where? When? Answer 'yes' or 'no' |
| | Comprehension questions | Interpreting extrapolating | State in your own word, explain, define, locate, select, indicate, summarise, outline, and match |
| High level thinking | Application questions | Applying information heard or read to new situations | Demonstrate how to, illustrate how to, show how to, apply, construct, and explain. What is ... used for? What would result? What would happen? |
| | Analysis questions | Breaking down into parts, relating parts to the whole | Distinguish, separate, outline, classify, contrast, compare, differentiate, what is the relationship between? What is the function? What motive? What conclusion? What is the main idea? |
| | Synthesis questions | Combining elements into a new pattern | Compose, combine, estimate, invent, choose, hypothesis, build, solve, design, and develop. What if? How would you test? What would you have done in this situation? What would happen if ... ? How can you improve ... ? How else would you ... ? |
| | Evaluation questions | Make a judgment of good, bad, right or wrong, according to some set of criteria, and state why. | Evaluate, rate, defend, dispute, decide which, select, judge, grade, verify, and choose why. Which is the best? Which is more important? Which do you think is more important? |

Adapted based on [36–38].

Examining the questions initiated by the child according to Bloom's taxonomy is a reliable method to measure the level of the child's thinking. Therefore, all questions in the child's data were identified and transcribed first and then analyzed according to the 'level of questions' (henceforth, in this article, abbreviated 'the LoQ') to determine his thinking level. Through these analyses, it is expected that research question 2 could be answered.

### 3.4. Reliability and Validation Process

The data analysis followed professional research practices [39,40] to ensure the reliability and validity of the results. The first author first transcribed the data, and the other authors checked the transcriptions. When a divergence arose, the researchers exchanged views until a consensus was reached. As a result, the inter-rater agreement was around 95%. Regarding assessing children's level of critical thinking based on the DR and the LoQ, we did not find any disagreements.

### 3.5. Ethical Concerns

The whole research was conducted in full compliance with relevant ethical guidelines. All the ethical-related issues, such as the study's purposes, procedures, confidentiality, anonymity, storage of data, and participants' rights, are clearly explained. Their consent was also sought before the study. The ethics approval number of this study is H14579.

## 4. Findings

Table 3 summarizes the main aspects of data collected in the research. These include data context, data length, and the total turn numbers. In total, 16 recordings have been collected in the research (8 weeks × 2 recordings). Each week's recording was divided into three files based on the data context. For example, Week 1 has Data No. 1 to 3 and Week 5 Data No. 13 to 15.

**Table 3.** The data profile.

| Week | Data No. | Data Context | Data Length | The Total Turn Numbers |
|---|---|---|---|---|
| | 1 | FC | 35 m 34 s | 150 |
| Week 1 | 2 | PC | 31 m 47 s | 277 |
| | 3 | SR | 37 m 36 s | 174 |
| | 4 | FC | 17 m 54 s | 71 |
| Week 2 | 5 | PC | 20 m 40 s | 63 |
| | 6 | SR | 39 m 27 s | 226 |
| | 7 | FC | 37 m 10 s | 141 |
| Week 3 | 8 | PC | 31 m 47 s | 99 |
| | 9 | SR | 35 m 36 s | 170 |
| | 10 | FC | 33 m 49 s | 79 |
| Week 4 | 11 | SR | 38 m 45 s | 84 |
| | 12 | PC | 28 m 52 s | 191 |
| | 13 | SR | 15 m 42 s | 87 |
| Week 5 | 14 | FC | 32 m 39 s | 67 |
| | 15 | PC | 38 m 09 s | 162 |
| | 16 | SR | 30 m 54 s | 89 |
| Week 6 | 17 | FC | 31 m 54 s | 55 |
| | 18 | PC | 42 m 13 s | 232 |
| | 19 | SR | 25 m 18 s | 98 |
| Week 7 | 20 | FC | 32 m 47 s | 108 |
| | 21 | PC | 20 m 50 s | 61 |
| | 22 | FC | 34 m 19 s | 74 |
| Week 8 | 23 | SR | 32 m 45 s | 100 |
| | 24 | PC | 45 m 51 s | 278 |

FC: free conversation, SR: story reading, PC: playing with the child, m: minutes, s: seconds).

### 4.1. The Exchange Analysis Based on The DR

This sub-section presents data analysis based on the DR. Below exemplifies each skill of critical thinking exchanges based on the six skills listed in 'the DR.'

4.1.1. Interpretation

1. Clarifying the meaning

After reading the story: 'Lisa wants a dog,' the mother conducted some after-reading discussions with the child.

Mother: *wo ye xiangyao yitiao gou.*

I also want have one dog.

'I also want a dog.'

Jay: *wo ye xiang yao yitiao gou.*

I also want have one dog.

'I also want a dog.'

Mother: *wo haixiang yang yizhi niao.*

I also want raise one bird.

'I also want to raise one bird.'

Jay: *wo ye xiangyang yizhi niao. Wo xiangyao yitiaogou wo xiangyao yizhi niao de yisi jiushi ne, jiaru wo henmang, ta yaowo peita qu sanbu, nawo fang yizhi niao, ta ting wo de hua, hui zhao huijia de lu de niao, ranhou nagou zhuizhe ta pao.*

I also want to raise one bird. I want have one dog I want have one bird's meaning is, if I am busy, he wants me company him take a walk, then I send the bird, it listens to my words, can find the home road of bird, then the dog chase it run.

'I also want to raise a bird, what I mean "I want a dog, and I want to raise a bird" is that if I am busy, but the dog wants me to take him out, I would like the bird to take him out. The bird is obedient to my order, and it also knows how to come home, then the dog can chase the bird to have a run.'

According to the DR, 'clarifying meaning' is 'to paraphrase or make explicit, through stipulation, description . . . ' [3] (p. 7). For example, the participant tries to explain what he meant and said, 'I want a dog and a bird' by saying 'what I mean . . . .' Therefore, the above exchanges can be categorized in the sub-skill of 'clarifying meaning' within the skill of 'interpretation.'

None of the child's utterances in the data belong to 'decoding significance' or 'categorization' in all transcription.

4.1.2. Analysis

1. Examining Ideas

After reading the story, 'Lisa wants a dog,' the mother tried some discussion with the child as in the following example:

Mother: *Lisa shi zenme youle ta ziji de gou a?*

Lisa is how have herself dog?

'How does Lisa get a dog?'

Jay: *tie zhitiao jie gou, danshi wo shi jueding mai gou.*

Stick a note borrow dog, but I decide buy dog.

'Left a note to borrow a dog, but I decide to buy a dog.'

Mother: *weishenme yao maigou ne?*

Why will buy dog?

'Why do you buy a dog?'

Jay: *maigou zheyangzi, zheyangzi neng ziji qi mingzi, danshi jiede naxie bieren de dou qi guo le.*

Buy dog in this way, in this way can myself name it, but borrow those other people dog already named.

'If I buy a dog, I can name the dog myself, but if I borrow a dog, it will already have been named'.

Mother: *hai you mei you bie de haochu le a?*

There is or not other benefit?

'Is there any other benefit we buy our dog rather than borrow one?'

Jay: *jiaru bierenjia yangde gou keyi zaichuangshang, womenjia kebuxing, na gou paodao women chuangshang zenmeban?*

If other people's dog can go to bed, in our home is not allowed, then the dog run to our bed, what do?

'If the previous owner of the dog allows the dog to get on to his bed, but in my home, it is not allowed to get on our bed, so if the dog gets on to our bed, what should we do with that?'

According to the DR, 'examining ideas' is 'to compare and contrast ideas . . . ' [3] (p. 7). In the above example, the child explains why he does not want to borrow a dog and provides his justification by comparing the advantages and disadvantages of buying versus borrowing a dog. Accordingly, this comparison can be categorized into the 'examining ideas' classification within the 'analysis' skill.

The child did not say anything regarding the 'identifying arguments' or 'analyzing arguments' in all transcription.

### 4.1.3. Evaluation

1. Assessing Claim

The mother was talking about a piece of news by saying:

Mother: *wo kan douyin shuo xini de haibian keneng yao you 5mi gao de hailing.*

I see TikTok say Sydney's seaside may have 5 m high wave.

'I saw a piece of news on TikTok saying that there probably will be a giant 5 m high wave along the coastline in Sydney.'

Jay: *douyin keneng shi jia de.*

TikTok could be fake.

'What TikTok said is possibly fake.'

Mother: *weishenme ni juede ta shi jia de?*

Why you think it is fake.

'Why do you think that is not true?'

Jay: *yinwei women dou meiyou ganshoudao yici haiyang chongji zheli de ganjue.*

Because we all not feel once wave hit here feeling.

'Because we have not felt anything that the hits from the waves.'

According to the DR, 'evaluation' is 'to assess the credibility of statements . . . ', and assessing claims is ' . . . to determine if a given claim is likely to be true or false based on what one knows . . . ' [3] (p. 8). When Jay heard what Mother said, he questioned the news's credibility by saying, 'What TikTok said is possibly fake.' After the mother's response, 'Why do you think that is not true?', the child provided reasons based on his personal experience. Although the reason he proposed is subjective, he demonstrated his ability to question the credibility and presented the reasons. Therefore, this thinking skill can be classified into the sub-skill of 'assessing claim' within the skill of 'evaluation'.

No relevant content about 'assessing arguments' was found during the analysis.

### 4.1.4. Inference

1. Conjecturing Alternatives

In this conversation, Father was talking about a video, 'Unsolved Mysteries.' Then, Jay proposed his hypotheses as to why the person died in that way.

Jay: *dagai shi you weiniu de daocai zai limian.*

Probably is have feed cow straw inside.

'It is probably that there was a pile of straw feeding cow inside the hole?'

Jay: *jiaru bierenjia yangde gou keyi zaichuangshang, womenjia kebuxing, na gou paodao women chuangshang zenmeban?*

Jay: *youkeneng shi ta yizoudao nage weizhi, nage weizhi hen qingbo, ranhou ne, buxiaoxin diaole xiaqu.*

Probably is he once walk that location, that place too thin, then, not careful fell down.

'It is probably that the place where he walked on is too thin, and then, he fell down inside.'

According to the DR, 'conjecturing alternatives' is 'to formulate multiple alternatives for resolving a problem, to postulate a series of suppositions regarding a question . . . ' [3] (p. 9). When Jay finished listening to his father's description of the 'Unsolved Mysteries' case, he presented two suppositions for the reasons behind it. Hence, the thinking skill applied by the child can be grouped into the sub-skill of 'conjecturing alternatives' within the skill of 'inference.'

2. Drawing Conclusions

The following is a conversation between Jay and his parents:

Father: *xianzai xuesheng gongzuochu mangzuo yi tuan, quanburen douzai zhunbei biye dianli he shuzi yingxin de gongzuo.*

Now student's department busy one chunk, all staff are preparing graduation ceremony and digital orientation work.

'The student department is bustling now, and all colleagues are preparing a graduation ceremony and also digital orientation program.'

Mother: *na zhiqian doushi ni yigeren zuo de, xianzai zhengge bumen dou zai zuo Zhejianshi.*

Then before all are you alone do then, now all department all do this work.

'You undertook these tasks on your own before, and now the whole department is doing these.'

Father: *shi a, tamen jiu zhidao wo zhiqian yige ren you duomang le.*

Yes, they then know I before alone have many busy.

'Yes, then they know how busy I was before.'

Mother: *haihao ni xianzai buzai zuo zhexie shiqing le.*

Good you now not do these works.

'It is good that you are no longer doing these things.'

Jay: *xianzai baba jiu shufu le.*

Now daddy is comfortable.

'Father is more comfortable now.'

According to the DR, 'drawing conclusions' is 'to apply appropriate modes of inference in determining what position, opinion or point of view one should take on a given matter or issue' [3] (p. 9). The parents talked about what happened to the previous department when the father left. The child listened and concluded that his dad is less stressed now, which is true.

No content regarding 'querying evidence' was found throughout the transcriptions.

4.1.5. Explanation

1. Stating Results

The following example happened when Jay was wearing a headset while watching a cartoon. Mother called him, but he did not answer. She took off his headset and asked:

Mother: *ni tingjian wo shuo hua le ma?*

You hear I speak?

'Did you hear me?'

Jay: *mei you.*

Not have.

'No.'

Then he added:

Jay: *mama,wo zhecai zhidao ni weishenme gongzuo de shihou buli wo le.*

Mom, I just know you why work time not reply me.

'Mom, I finally understand why you ignore me when you work.'

Mother: *en.*

Uh-huh.

'Uh-huh.'

Jay: *weishenme wo yao zheme shuo ne? Yinwei jintian wo daizhe erji, wo jiu tingbujian ni de shengyin a.*

Why I want this way say? Because today I wear headset, I then cannot listen your voice.

'I said this because I wore a headset today, and I could not hear you.'

According to the DR, 'stating result' is 'to state one's reasons for holding a given view' [3] (p. 9). Jay first stated that he understood why his mother could not hear him when she was working. Then, he proposed a question reflecting why he said that and then presented why he held that view. Therefore, the child adopted the sub-skill of 'stating result' within the skill of 'explanation'.

2. Presenting Arguments

Jay asked his mother where his monster car (a toy car) was.

Son: *deviance youmeiyou dai?*

Monster car have or not have bring.

'Did I bring my Monster car?'

Mother: *zai ni shubao li a.*

At your bag inside.

'It is in your bag.'

Jay: *oh.*

Oh.

'Oh.'

Mother: *ni dou wangji ni beizhe shubao le ba?*

You all forget you carry bag?

'Did you forget that you are carrying your bag on your back?'

Jay: *en, taishufu le, beizhe shubao.*

Yes, too comfortable, carrying bag.

'Yes, the bag is so comfortable to carry.'

Mother: *en,ni ganjue budao ni beizhe tamen zai.*

Yes, you feel not you carry them.

'Uh-huh, so, you couldn't feel you were carrying it.'

Jay: *en, ta ye tai qing le.*

Yes, it also too light.

'Uh-huh, and it also so light.'

In line with the DR, 'presenting argument' is 'to give reasons for accepting some claim' [3] (p. 9). The child expressed two reasons for accepting the mother's claim: 'you forgot you are carrying your bag on your back.' This reflects that the child uses the sub-skill of 'presenting argument' within the skill of 'explanation.'

No relevant content was found in the data regarding the 'justifying procedures' throughout the analysis.

### 4.1.6. Self-Regulation

1. Self-Examination

Jay lost his toy, so his mother tried to educate him to take care of his toy, as in the following example:

Mother: *ni weishenme bu aixi ni ziji de wanju, zheshi baba mama yong qian gei ni maihuilai de, yeshi ni ziji qinzi tiaoxuan de.*

You why not cherish yourself toy, this is daddy mom spend money give you buy back, also yourself personal select.

'Why didn't you look after your toy? Parents spent money and bought it for you, and you chose it yourself.'

Jay: *mama zongshi yao mawo, dangranla, yinwei niba guidao gaobujian la.*

Mom always blame me, of course, because you let track lost.

'Mom always blames me, of course, because you lost a track.'

According to the DR, 'self-examination' is 'to reflect on one's motivations, values, attitudes and interests to determine that one has endeavored to be unbiased, fair-minded, thorough, objective' [3] (p. 10). For example, Jay first presented that 'Mom always blames me'; however, he soon changed his attitude and reflected on what he said by talking to himself in the second person 'ni (you).' This is an act of self-reflection on his own words and behavior where he finally reaches an unbiased and fair-minded view. Therefore, this thinking can be sorted into the sub-skill of 'self-examination' within the skill of 'self-regulation'.

Through examining all the transcripts, the contents related to 'self-correction' have not been identified.

Furthermore, we found that the demonstration of CT can be divided into two classifications: 'spontaneous one' (SO) and 'triggered one' (TO). 'SO' is a spontaneous statement without guidance or stimulation as in (1), while 'TO' is a statement with guidance or stimulation, as in (2).

(1) After watching a short video on TikTok about a war, Jay stated:

*fashe yuanzidan de guojia shi buhao de, danshi weishenme nage guojia yao fashe gei tamen ne? yinwei tamen de guojia xian long le nage haoren de guojia yiba, ranhou ne, nage haoren de guojia fankang huiqu,ranhou cai kaishi zhanzheng de.*

Launch nuclear bomb country is not good, but why that country want launch to them? Because their country first attacks the good people country one time, then, the good people country fight back, then begin the war.

'The country who launched the nuclear bomb was not right, but why did the country do this? It was because the other country attacked this country first, and then this country was just fighting back. This is how the war began.'

Based on the DR, a sub-skill of 'stating result' is 'to state one's reasons for holding a given view' [3] (p. 10). The child first proposed a statement and followed it by a question to reflect on why he had said this; he then expressed why he held this view. Therefore, the skill demonstrated here by the child is the sub-skill of 'stating result' within the skill of 'explanation'. Moreover, this was his spontaneous statement; therefore, applying the skills in this way should be labeled as 'SO.'

In (2), Jay wanted water from his father, which was ignored because the father was playing on TikTok. Then, Jay said:

(2) Jay: *mama yaoba baba shouji shang de douyin shanchu le.*

Mom. Want let daddy's phone's TikTok delete.

'Mom, the TikTok on father's mobile phone should be deleted.'

Mother: *weishenme?*

Why?

'Why?'

Jay: yinwei ta zongshi kan douyin.

Because he always looks TikTok.

'Because he always plays TikTok.'

Mother: kandouyin, ranhou ne? you naxie buhao de cai shanchu a, ruguo kan douyin henhao jiu buyao shanchu a!

Look at TikTok, then? Have what not good then delete, if look at TikTok is good, then not delete.

'Playing TikTok, and then? What are the drawbacks leading to uninstall TikTok? If it is good to play TikTok, then we do not need to uninstall.'

Jay: *kan dongyin rangta buting ren jianghua.*

Look TikTok let him not listen people speak.

'He ignored our talk because of playing TikTok.'

Mother: *en, henhao, zheshi yige buhao de quedian, hai you meiyou?*

Yes, good, this is one not good disadvantage, have not have?

'Uh-huh, this is one disadvantage, anything else?'

Jay: *haiyou weile kan douyin, dou bu haohao chifan.*

And in order to look TikTok, not good good eat meal.

'He does not eat well because of playing TikTok.'

Mother: *en, zhege geng buhao,yingxiang chifan de dou bukeyi, hai youmeiyou?*

Yes, this is more not good, affect eat meal all not allowed, have not have?

'Uh-huh, this is a worse drawback; anything else?'

Jay, *haiyou, ta kan douyin buhui zuofan.*

More, he looks TikTok not able to do meal.

'Moreover, he does not cook because of playing TikTok.'

Mother: *diqueshi, kan douyin you meiyou hao de defang?*

Indeed, look TikTok have not have good points?

'Indeed, but is there any benefit of playing TikTok?'

Jay: *ta kan douyin ye neng xiuxi yihui.*

He looks TikTok also can rest one while.

'He can get rest by playing TikTok for a while.'

Mother: *henhao, hai youmeiyou youdian le?*

Very good, have not have advantage?

'Great! Any other benefit?'

Jay: *meiyou le.*

Not have.

'No.'

In (2), Jay states three reasons why he believed TikTok should be uninstalled from his dad's mobile phone with the help of the trigger of the mother: 'What are some drawbacks which would lead to uninstalling TikTok?', and with the mother's help, 'Indeed, but are there any positive benefits of playing TikTok?', he can also posit some evaluations for pros and cons of playing TikTok. Thus, the way he applies evaluation should be labeled as 'TO.' Parents' use of questions prompted the child's thinking and expanded his critical thinking capacity. They played the role of scaffolding in the child's zone of proximal development [41].

As shown in Table 4, a total of 85 instances can be considered as applying cognitive skills by the child. The most frequent skill he applied is 'inference', which accounts for 32.9% (i.e., 28 instances out of 85 in total). Interestingly, two pairs of skills are in the same proportion. These are 'interpretation' and 'self- regulation' (both are 12.9%) and 'analyses' and 'evaluation' (both are 9.4%). This latter pair of skills were the least frequently used. Moreover, in terms of two indicators, the percentage of 'SO' outweighs the 'TO', 56.4% and 43.5%, respectively.

**Table 4.** Frequencies of the child's critical thinking based on the Delphi report.

|   | Skill | Sub-Skills Categorization | Amounts of Appearances | SO | TO | Total of Incidence and Percentages |
|---|---|---|---|---|---|---|
| 1 | Interpretation | Categorization | / | / | / | 11 (12.9%) |
|   |   | Decoding significance | / | / | / |   |
|   |   | Clarifying meaning | 11 | 9 | 2 |   |
| 2 | Analysis | Examining ideas | 8 | 3 | 5 | 8 (9.4%) |
|   |   | Identifying arguments | / | / | / |   |
|   |   | Analyzing arguments | / | / | / |   |
| 3 | Evaluation | Assessing claims | 1 | 1 | / | 8 (9.4%) |
|   |   | Assessing arguments | 7 | 7 | / |   |
|   |   | Querying evidence | / | / | / |   |
| 4 | Inference | Conjecturing alternatives | 25 | 15 | 10 | 28 (32.9%) |
|   |   | Drawing conclusions | 3 | 3 | / |   |
|   |   | Stating results | 19 | 3 | 16 |   |
| 5 | Explanation | Justifying procedures | / | / | / | 19 (22.4%) |
|   |   | Presenting arguments | / | / | / |   |
| 6 | Self-Regulation | Self-examination | 7 | 5 | 2 | 11 (12.9%) |
|   |   | Self-correction | 4 | 2 | 2 |   |
|   |   | Total | 85 | 48 (56.4%) | 37 (43.5%) | 85 (100%) |

SO = spontaneous one, TO = triggered one'.

### 4.2. The Question Analysis Based on the 'Level of Questions'

The child produced 348 questions in total. All these questions were analyzed following the 'the LoQ.' Question types are exemplified below:

#### 4.2.1. Knowledge Questions

According to 'the LoQ,' 'knowledge questions' are 'eliciting factual answers, testing recall and recognition of information.' The questions below ask the basic information on what and where. Therefore, these questions are classified as 'knowledge questions.'

Examples of questions in this category are presented below:

(1) *qiqiu fangzai na?*

Balloon put where?

'Where is balloon?'

(2) *Zai jiaoshi li wanle shenme?*

In classroom play what?

'What did you play in the classroom?'

### 4.2.2. Comprehension Questions

According to 'the LoQ,' a 'comprehension question' is 'interpreting' and 'extrapolating,' which require the interlocutor to make explanations. In the examples *(3)* and *(4)*, the child asked the parents to explain; therefore, they are categorized into 'comprehension questions.'
Examples of questions in this category are presented below:

(3) *Rezhanglengsuo shi shenme yisi?*

Hot expansion cold contraction is what meaning?

'What does 'thermal expansion and contraction' mean?'

(4) *Shuini weishenme hui gan?*

Concrete why can dry?

'Why does concrete become dry?'

### 4.2.3. Application Questions

The child asked questions using 'how' to find the method of turning off the clock *(5)* and printing *(6)*. Thus, these questions are defined as 'application questions.'
Examples of questions in this category are presented below:

(5) *Naoling zenme guan?*

Clock how turn off?

'How do I turn off the alarm clock?'

(6) *Zenme dayin?*

How print?

'How to print?'

### 4.2.4. Analysis Questions

According to 'the LoQ,' an 'analysis question' relates to distinguishing one from another; for example: *(7)*, the bad guy between the good guy, and *(8)*, a cute child between a bad child. Therefore, these are categorized into 'analysis questions.'

(7) *Shei shi huairen ne?*

Who is bad people?

'Who should be considered as a bad guy?'

(8) *Daodi shi keai de nanhai haishi huai xiaohai a?*

On earth is cute boy or bad kid?

'What on earth is he a cute child or a bad child?'

### 4.2.5. Synthesis Questions

According to 'the LoQ,' a 'synthesis question' is 'combining elements into a new pattern' using the key questions words 'what if,' 'what would happen if,' and so on. In examples (9) and (10), the child proposed a hypothesis using the questions words 'what if' and 'what would happen if. According to the Processability theory, a psycholinguistic model of language acquisition, Pienemann emphasized that this sort of question construction, involving complex clauses, is placed at a higher stage of language acquisition because these involve higher demand on language processing [42] and a higher cognitive level [43].

(9) *Na kai yige tianchuang ne?*

Then open one skylight?

'What would happen if there was a skylight?'

(10) *Wanyi ta zai women jia la baba zenme ban?*

In case it at our home do poop how do?

'What if it (a dog) poops at our house?'

4.2.6. Evaluation Questions

According to 'the LoQ,' the 'evaluation questions' are for 'making a judgment' using the question word 'why.' The examples (11) and (12) below manifest that the child made a statement first and asked a 'why' question. Then, he stated the reason for his judgments.

(11) *fashe yuanzidan de guojia shi buhao de, danshi weishenme nage guojia yao fashe gei tamen ne? yinwei tamen de guojia xian long le nage haoren de guojia yiba, ranhou ne, nage haoren de guojia fankang huiqu, ranhou cai kaishi zhanzheng de.*

Launch nuclear bomb country is not good, but why that country want launch to them? Because their country first attacks the good people country one time, then, the good people country fight back, then begin the war.

'The country who launched the nuclear bomb was not right, but why did the country do this? Because the other country hit this country first, and then this country just fought back, and then the war began.'

(12) *mama,wo zhecai zhidao ni weishenme gongzuo de shihou buli wo le. weishenme wo yao zheme shuo ne? Yinwei jintian wo daizhe erji, wo jiu tingbujian ni de shengyin a.*

Mom, I just know you why work time not reply me. why I want this way say? Because today I wear headset, I then cannot listen your voice

'Mom, I finally understand why you ignore me when you work. Why I said this is because I wore a headset today, and I cannot hear you.'

Moreover, by examining the total questions, we found that the questions can also be classified into two categories: 'independent questions (IQs)' and 'continuous question (CQs).' IQs are isolated questions without further questions when the child obtains the answer. In contrast, 'CQs' mean that the child issues further questions for a deeper understanding of the same topic after the original question. 'CQs' are classified as a higher level of thinking than 'IQs', since the 'CQs' require more logic and further thinking processes. The following are the examples of 'IQs' as in (13) and 'CQs' as in (14).

1, IQs

This question was asked when the child saw a concrete mixer parking along the road, and he did not ask any further questions about it.

(13) *Shuini weishenme hui gan?*

Concrete why can dry?

'Why does concrete become dry?'

2, CQs

The following conversation happened during dinner when one dish was the meretrix. The participant asked several questions about the clam, meretrix, and freshwater mussel, trying to understand their similarities and differences. The child continued asking questions about the topic until achieving his understanding.

(14) *Huajia shi ta de haizi ma?*

Clam is his kids?

'Are clam their (meretrix) children?'

*Zhexie zhengzhujia shibushi keyi ba shitou biancheng zhengzhu a*

These meretrix is not is can let stone change into pearl?

'Does these meretrix can change stone into a pearl?'

*Bang shi shenme?*

Freshwater mussel is what?

'What is a freshwater mussel?'

*Weishenme zhexie beike zhuzhe jiu neng ba ta dakai ne?*

Why those meretrix boiling then can let then open?

'Why do these meretrices open when boiling them?'

*Rezhang lengsuo shi shenme yisi?*

Hot expansion cold contraction is what meaning?

'What does thermal expansion and contraction mean?'

The 348 questions analyzed are summarized in Table 5.

As can be seen from Table 5, among 348 questions, 330 cases (95%) were classified into low-level thinking, and the rest, 18 cases (5%), were sorted into high-level thinking. Knowledge questions account for the most percentage within low-level thinking, around 74%, while within the high-level thinking, synthesis questions were the highest, making up 0.8%. Moreover, in terms of 'IQ' and' CQ', 'IQ' is much higher than 'CQ', almost double that of 'CQ.'

**Table 5.** Thinking level of questions proposed by the child.

|  | Level of Questions | Total Number of Incidences (Percentages) | | IQ | CQ | Thinking Levels |
|---|---|---|---|---|---|---|
| 6 | Evaluation questions | 2 (0.5%) | | 2 (100%) | / | High level thinking HLT = 18 (5%) |
| 5 | Synthesis questions | 3 (0.8%) | | 3 (100%) | / | |
| 4 | Analysis questions | 2 (0.5%) | | 1 (50%) | 1 (50%) | |
| 3 | Application questions | 11 (3.2%) | | 11 (100%) | / | Low level thinking LLT = 330 (95%) |
| 2 | Comprehension questions | 72 (20.6%) | | 48 (66.6%) | 24 (33.3%) | |
| 1 | Knowledge questions | Yes-no Qs | 175 (50%) | 124 (70.9%) | 51 (29.1%) | |
| | | Wh-Qs | 83 (23.9%) | 35 (42.2%) | 48 (57.8%) | |
| | | Total: 348 questions | | 224 (64.3%) | 124 (35.6%) | 348 (100%) |

IQ = independent questions, CQ = continuous questions.

## 5. Discussion

The results indicate that the child demonstrated critical thinking in out-of-class situations based on the DR and the LoQ. This proves that young children can show signs of possessing critical thinking in a home setting, supporting that this psychological process begins in the early years [22,26,27,29,30]. Moreover, it also supports the idea that young children are capable of evaluating the reliability of the information in the light of their epistemic knowledge [44]. Finally, it further adds empirical evidence to previous research [45–47], suggesting that 'young children are not wholly credulous of information but rather are selective [44].'

Moreover, one point worth noting is that children's cognitive skills were stimulated under the intervention of teachers' classroom activities in previous research [20,26,27,29,30]. Our study also found that the child could express his critical thought naturally without any intervention. In fact, the child demonstrates his intellectual process more spontaneously, which accounts for 56.4% of interactions, compared to the triggered way (43.5%). As put forward before, the SO indicator reflects children's critical thinking more naturally, rather than in an 'artificial' way, supporting the idea that young children are active in acquiring information rather than being passive recipients of knowledge [44]. This may be driven by the psychological nature of young children, inborn and natural curiosity [7]. This unique advantage may provide them with an open-mind attitude rather than a stereotypical one toward the outside world around them. Another significant reason may be the positive family background, such as the friendly home environment and supportive parental involvement. As pointed out by Vygotsky in the well-known Sociocultural theory, the cognitive development of early childhood is advanced through social interaction with other people, particularly those who are more skilled [41]. Therefore, the interactions between the child and parents play a fundamental role in the child's cognition development.

The current study empirically supports the Sociocultural theory and provides evidence to explain why Vygotsky emphasizes children's social interaction, especially between adults and children [48,49].

Furthermore, the ability of children to propose questions to gather information constitutes an efficient mechanism for cognitive development [50]. It demonstrates that children are active learners who seek to better understand the world around them. Out of 348 questions evaluated, low-level thinking questions account for 95%, while high thinking levels are 5%. This means that the child's thinking level reflected by the questions proposed has much room for future improvement. A high percent of low-level thinking questions may be due to the need for cognitive development at the child's age when more content knowledge is needed to prepare the child to propose questions with high-level thinking. It is worth noting that the child's CQs occupy more than one-third of the total number of questions. The CQs indicate that the child is actively engaged and highly motivated to clear what Piaget proposed as a 'disequilibrium situation' [10]. This means that the child is equipped with an inner critical thinking quality, which the current researchers here would term as the 'innate instinct' of critical thinking.

One contribution of the current research is that, in terms of research context, to our knowledge, most studies of young children's cognitive development are conducted within a classroom context. Yet, few studies have been conducted in out-of-class settings. The current study could indicate that researchers have placed too much attention on classroom contexts rather than considering the outside classroom settings when trying to acquire a comprehensive understanding of young children's critical thinking.

Secondly, this study added much-needed methodological tools to study young children's critical thinking in out-of-class settings. In terms of the research method, Ennis has emphasized that natural observation, such as a case study, has been a reliable way to examine young children's critical thinking below fourth grade [33]. However, most previous studies applied prepared and designed classroom activities for research. This current research demonstrates how to investigate children's critical thinking in a real natural way in the form of case study. Moreover, as proposed by Facione [4], 'the Delphi Report is intended as a guide to curriculum development and critical thinking assessment.' Many critical thinking assessments are motivated by this rich framework and take it as conceptual bases, such as the California Critical Thinking Skills Test [3] and The California Critical Thinking Dispositions Inventory [51]. The current study tries to practically apply it as a research measurement to investigate critical thinking, which made an effort to extend the Delphi Report application in a qualitative practice rather than a quantitative way.

Thirdly, although there are many reliable assessments on critical thinking, there is no critical thinking assessment available now for children below the fourth grade [33]. Very few studies try to measure the level of critical thinking of children below the fourth grade qualitatively. The current case study makes the first attempt to measure a young child's critical thinking in both qualitative and quantitative ways. The current study discovers two reliable indicators. The first is the 'spontaneous one', which could reflect the natural situation of the child's 'inborn' critical thinking. The second indicator is 'continuous questions'. As illustrated by Elder and Paul, 'Only when an answer generates a further question does thought continue its life. This is why you think and learn only when you have questions' [52] (p. 36). The continuous questions here are the further questions. It demonstrates children's persistence in seeking information, which reflects two sprits of critical thinking: the 'diligence in seeking relevant information' and 'focused in inquiry' [4] (p. 3); it further shows the core of critical reflection [53].

## 6. Conclusions, Limitations, and Implications

This case study examined a Mandarin-speaking child's critical thinking in an out-of-class, home setting. The research applied a qualitative analysis of the child's naturalistic speech productions to investigate two research questions. Additionally, quantitative analyses were conducted regarding the child's skills of critical thinking categorization and level

of questions. We posited two research questions. The first question asked whether the preschool child showed signs of possessing critical thinking in an out-of-class home setting. The results show that the child does demonstrate critical thinking. As for the second question, the level of critical thinking has been reflected in two indicators: 'SO' (spontaneous one) and 'CQ' (continuous questions). We specifically found that while the child exhibited all six categories of critical thinking skills based on the Delphi report, his level of questions was "low level". Therefore, we identified that the child's development area now is "high level" questions, and educators, both teachers and parents, are encouraged to help the child develop high-level questions, including synthesis and application questions.

Up to now, there has been very little normative data to measure children's typical critical thinking level below fourth grade. This current research is a first step in involving a new domain of children's critical thinking research. Moreover, the current research also provides a reliable method to measure critical thinking in children below grade four.

This study, however, has some limitations. Firstly, the current study is a case study.

Although the generalizability has been pointed out as a limitation of the case study method, scholars state that the case study is still a desirable way to gather detailed and rich qualitative data to uncover complex aspects of human thinking, behavior [54], and language [55]. Second, it is worth noting that the level of children's critical thinking when interacting with parents may vary with peers or teachers. Therefore, future studies could investigate exchanges between children and peers and between children and teachers to acquire a comprehensive understanding of children's critical thinking. As well as this, future research could be conducted to investigate children's 'CQ' at different ages (from 3 years to 9 years) to develop a form of children's critical thinking stages, which could contribute to the gap in recent critical thinking assessment.

Nevertheless, the findings of the current research have some implications. Firstly, language is a cultural tool [41] by which people can communicate, think, and solve problems. Therefore, to cultivate children's critical thinking, emphasis on improving children's language ability should be placed. This should be the top priority when trying to improve children's critical thinking in any programs, most of which pay more attention to cultivating children on how to think rather than how to express themselves by language.

Secondly, the interaction in the sociocultural context is vital for fostering children's critical thinking. The current research suggests that parents' supportive involvement in conversation and a friendly home environment is valuable and essential to children's cognitive maturity. Therefore, parents are encouraged to be active and supportive when interacting with children in free conversation, story reading, and playing games. Moreover, it is expected that the current study could be an educational or clinical indicator for researchers, educators, and parents to understand the children's normative cognitive development in critical thinking at the stage.

**Author Contributions:** Conceptualization, X.S.; Methodology, X.S.; Data curation, X.S.; Writing—original draft preparation, X.S.; Writing—review and editing, X.S., R.Q., S.K. and H.L. All authors have read and agreed to the published version of the manuscript.

**Funding:** This research received no external funding.

**Institutional Review Board Statement:** The study was conducted in accordance with the National Statement on Ethical Conduct in Human Research 2007 (Updated 2018), Australia, and approved by the Western Sydney University Human Research Ethics Committee (HREC Approval Number: H14579 and date of approval: 17 November 2021).

**Informed Consent Statement:** Informed consent was obtained from all subjects involved in the study.

**Data Availability Statement:** The data are not publicly available due to ethical and privacy restrictions, e.g., their containing information that could compromise the privacy of research participants.

**Acknowledgments:** The authors would also like to thank the parents and the child for their time participating in this study.

**Conflicts of Interest:** The authors declare no conflict of interest.

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
