# Peer review of "Early Critical Thinking in a Mandarin-Speaking Child: An Exploratory Case Study"

_education, doi:10.3390/educsci12020126_

Round 1

Reviewer 1 Report

Overall, the article presents an examination of the critical thinking of a Mandarin-speaking child aged 5 years and 8 months in the out-of-class context, which is interesting for educational science. Educators develop critical thinking at all stages of the educational process; this is a really important aspect worth paying attention to.

The manuscript is scientifically sound and relevant for the field. The article is well-grounded in research design. Since the article is about children, it is important that ethical-related issues are described in the manuscript. The treatment of concepts and theories is precise. The conclusions are consistent with the evidence and arguments presented. There is a relation between the title, the problem, the aim, the theoretical framework, the methods, the findings, the discussion, and the conclusions. The contribution of the authors is distinguishable from other academic texts on the subject and is useful for the field in a new context.

The authors are encouraged to consider the following improvements:

  • Each sentence in the first paragraph contains the phrase "critical thinking" – from a literary point of view (including scientific literature), in some sentences, it might be possible to replace this phrase, for example, with "intellectual / psychological process".
  • The article provides clear and robust arguments for studying critical thinking in an out-of-class setting, but the importance of examining the critical thinking of a Mandarin-speaking child is not supported by the arguments and references.
  • Although the article examines the psychological process in the linguistic context, there are not enough references to psycholinguistic literature and psycholinguistic findings accordingly.
  • The mention of The Delphi Report appears too abruptly in the article, it would be better if the readers, by the time this research element appears, were ready to agree on the need to apply it.
  • “To ensure data reliability, researchers first transcribed these recordings and then examined transcripts according to ‘The DR’ and ‘The LoQ,’ When a divergence arose, the researchers exchanged views until a consensus was reached” - the statistical reliability of the results requires a more detailed justification.

Reviewer 2 Report

In the work, the literature review is done quite superficially.
There is no reference to the latest studies and achievements in this subject.
Author writes that: "This study adopts a case study, utilizing a naturalistic
environment, observation, and language samples" -
the author does not explain exactly what
the observation is and what are the language samples.
Very poor description of the methodology.
The author also does not provide the year in which the study was conducted.
In conclusion, I also propose to refer to the benefits of information
for parents about how critical thinking looks like in children.

Reviewer 3 Report

The paper presents a study about critical thinking with a preschool child. It is well written and makes clear what was developed.

I/we think the authors could  provide further discussion about the Delphi Report as it is used as a basis for the analysis of the speeches of the study carried out.

I/we could not understand the context of conversation during the topic 2 (Drawing conclusions). Please provide more information about this conversation.

It was a bit difficult to understand Table 2 - Level of questions. I think that using a grid with lines can help us to better understand what item item belongs to each thinking level.

Why the lines of the article are numbered (right side of the paper)?
